# A Closer Look at Opioid-Induced Adrenal Insufficiency: A Narrative Review

**DOI:** 10.3390/ijms24054575

**Published:** 2023-02-26

**Authors:** Flaminia Coluzzi, Jo Ann K. LeQuang, Salvatore Sciacchitano, Maria Sole Scerpa, Monica Rocco, Joseph Pergolizzi

**Affiliations:** 1Department Medical and Surgical Sciences and Biotechnologies, Sapienza University of Rome, Polo Pontino, 04100 Latina, Italy; 2Unit of Anaesthesia, Intensive Care, and Pain Medicine, Sant’Andrea University Hospital, 00189 Rome, Italy; 3NEMA Research, Inc., Naples, FL 34108, USA; 4Department of Clinical and Molecular Medicine, Sapienza University of Rome, 00189 Rome, Italy; 5Laboratory of Biomedical Research, Niccolò Cusano University Foundation, 00166 Rome, Italy; 6Department of Surgical and Medical Science and Translational Medicine, Sapienza University of Rome, 00189 Rome, Italy

**Keywords:** opioids, adrenal insufficiency, chronic pain, endocrine, adrenal crisis, pituitary, hypothalamic-pituitary-adrenal axis, opioid use disorder

## Abstract

Among several opioid-associated endocrinopathies, opioid-associated adrenal insufficiency (OIAI) is both common and not well understood by most clinicians, particularly those outside of endocrine specialization. OIAI is secondary to long-term opioid use and differs from primary adrenal insufficiency. Beyond chronic opioid use, risk factors for OIAI are not well known. OIAI can be diagnosed by a variety of tests, such as the morning cortisol test, but cutoff values are not well established and it is estimated that only about 10% of patients with OIAI will ever be properly diagnosed. This may be dangerous, as OIAI can lead to a potentially life-threatening adrenal crisis. OIAI can be treated and for patients who must continue opioid therapy, it can be clinically managed. OIAI resolves with opioid cessation. Better guidance for diagnosis and treatment is urgently needed, particularly in light of the fact that 5% of the United States population has a prescription for chronic opioid therapy.

## 1. Introduction

Opioids have been and remain among the most prescribed analgesics in the developed world, despite concerns about tolerance, use disorder, diversion, and adverse effects [1]. Approximately 9% to 29% of chronic opioid therapy patients develop opioid-induced adrenal insufficiency (OIAI), which is challenging to diagnose and may be treatment limiting [2,3]. OIAI symptoms include nausea, vomiting, lethargy, weight loss, dizziness, and muscle aches and pain. Some of these diffuse symptoms may be difficult to assess as they can overlap with the patient’s prescribing indication [2]. OIAI can occur with any opioid and despite its prevalence, has not been the subject of great academic interest [2]. Note that OIAI occurs secondary to prolonged exposure to opioids and differs in many ways from primary adrenal insufficiency.

The hypothalamic-pituitary-adrenal (HPA) axis describes a complex network of interacting endocrine pathways in the brain, which allow for communications among the hypothalamus gland, the anterior pituitary gland, and the adrenal system [4]. The HPA axis regulates the body’s physiological responses to stress as well as regulating the appropriate release of hormones for sexual development. When it is functioning optimally, the HPA allows for physiological homeostasis, but disruptions and imbalances can trigger alterations in neuropeptide and neurotransmitter synthesis, some of which may have long-lasting consequences [4]. HPA dysfunction is common in those in long-term opioid therapy [5].

Despite the opioid epidemic, chronic opioid use in large subpopulations, and the widespread use of medication-assisted treatments for opioid rehabilitation which require prolonged exposure to opioids, opioid-induced endocrinopathies remain under-diagnosed and underappreciated [6]. The aim of this article is to provide a narrative overview of what is known about OIAI and its treatment.

## 2. Methods

This is a narrative review. The search term “opioid-induced adrenal insufficiency” was searched in December 2022 in PubMed with no delimiters and yielded 17 results. The keywords “opioid and adrenal insufficiency” with no delimiters yielded 86 results. The bibliographies of these articles were searched. The Cochrane Library database was searched for that same search time (0 results) and “adrenal insufficiency” (0 results). There were two results for “endocrinopathy” in the Cochrane database but neither related to opioids or adrenal insufficiency. Google Scholar was searched as well. There were some duplications among the results. This is a narrative review and there is limited published literature. There are no published clinical trials or randomized clinical trials on opioid-induced adrenal insufficiency in the literature. It should be noted that almost all of the peer-reviewed material published on this topic has appeared in the last five years.

## 3. Results

Located throughout the body, the main G-protein-coupled opioid receptors, mu-opioid receptors (MOR), delta opioid receptors (DOR), and kappa opioid receptors (KOR), respond to endogenous and exogenous opioids and suppress the HPA axis. Not unsurprisingly, there is scant evidence investigating HPA-axis suppression with only a few studies, which use different definitions, opioids, routes of administration, and doses, as well as different patient populations and markedly different study designs [2]. Different definitions make epidemiologic reference difficult. When HPA axis suppression is defined as a basal cortisol level of <5 µg/dL, then its incidence is 9.2% for those taking intrathecal morphine or hydromorphone, but if an insulin tolerance test is used to define HPA axis suppression, the rate of HPA axis suppression in the same population is 15%. Furthermore, if a 24 h cortisol-free urine assay below the reference range (20 to 90 µg/24 h) is used to define HPA axis suppression, then the rate is 20% [7].

The HPA axis may be thought of as a complex network of neuroendocrine pathways and feedback loops with the purpose of maintaining physiologic homeostasis [4]. The HPA axis is bounded by the hypothalamus, the anterior pituitary gland, and the adrenal system. The HPA axis develops in the fetus, but it is not until puberty that it develops fully under the influence of circulating gonadal hormones. Derangement in this system can alter neuropeptides, change neurotransmitter synthesis in the central nervous system, and disrupt glucocorticoid synthesis in the periphery [4]. HPA axis dysfunction can result not only in changes in the neuroendocrine system, but it can also cause changes in behavior, autonomic function, and metabolism [4]. Short-term exposure to opioids can suppress the HPA axis. Paradoxically, brief opioid exposure has the opposite effect in many animal studies and tends to increase corticotropin and glucocorticoids [2], making it challenging to use animal studies to investigate HPA axial suppression caused by brief opioid use. On the other hand, long-term exposure to opioids in both humans and animals has been associated with suppression of the HPA axis [2]. Studying HPA axial suppression is further confounded by marked intersubject variations. The mechanisms to explain why and how prolonged exposure to opioids can suppress the HPA axis include genetic polymorphisms of the opioid receptors, genetic variations in interleukin beta (ILβ), as well as differences that trace back to study design or methodology [2].

Peripheral (primary) and central (secondary and tertiary) forms of adrenal insufficiency occur because of direct adrenal gland disorders, such as rare genetic conditions or autoimmune diseases and secondary to ACTH deficiency, or by suppression of CRH by exogenous glucocorticoids or other medications, such as opioids, respectively [8,9]. In serum, cortisol has a 90-min half-life and the relative or absolute absence of endogenous cortisol, an important glucocorticoid that promotes homeostasis, has numerous consequences, including increased pro-inflammatory cytokines, altered populations of immune cells, and vascular consequences that can cause vasodilatation and profound, even life-threatening hypotension [10].

The deterioration of HPA axis communication can lead to any number of conditions, including OIAI. This is a secondary/tertiary form of adrenal insufficiency because it is a direct consequence of the inhibitory activity of opioids on the HPA axis rather than damage to the adrenal cortex itself [2]. See Figure 1.

## 4. Epidemiology of OIAI

Most opioids are prescribed to manage moderate-to-severe acute pain, but opioids are also prescribed long term to patients with cancer pain or chronic non-cancer pain [11]. In fact, over 5% of the United States population has a prescription for chronic opioid therapy [12]. Chronic opioid therapy has been associated with OIAI, but specific risk factors (agent, dose, duration, route of administration) have not been conclusively determined [13].

Since it was thought that the prevalence of OIAI was underestimated, a prospective study at a pain center was designed to evaluate adults taking opioids for ≥90 days [14]. Morning cortisol levels were evaluated at baseline. Based on endocrine testing, 5% of the patient population (n = 162) had OIAI and those patients with diagnosed OIAI were taking more morphine milligram equivalents (MME) per day than those without OIAI [14].

## 5. Risk Factors for OIAI

While suppression of the HPA axis is known to be a factor in the development of OIAI, the risk factors for HPA-axis suppression in chronic opioid therapy patients are not known [2]. Another opioid-associated endocrinopathy, opioid-induced androgen deficiency (OIAD), has been linked to high doses of long-acting opioids [15], but there is no conclusive evidence this is the case with OIAI [2]. It is not known if age, sex, or genetic polymorphisms contribute to OIAI and no study has yet evaluated whether OIAI is more or less associated with specific opioids or specific routes of administration [13]. It has been suggested that long-acting opioids may confer a greater risk of OIAI than other opioid formulations, but there is limited evidence to support this [16].

It may be that long-term exposure to opioids is the strongest risk factor for secondary adrenal insufficiency [17]. Opioid use disorder (OUD) may lead to OIAI, as a case report in the literature found, and the opioid epidemic may be at the root of many cases of OIAI [18]. A 28-year-old female presented at the emergency department with palpitations, malaise, hot flashes, and lightheadedness [19]. She had a history of hypertension, carpal tunnel syndrome, depression, and anxiety for which she was being treated. She had used marijuana regularly for the past 13 years and in the past five months had started using synthetic marijuana as well. She was prescribed 15 mg of oxycodone twice daily for her carpal tunnel syndrome, she was taking 60 mg/day by mouth and 60 mg intranasally per day for the past two years (120 mg/day). A random cortisol test and cosyntropin suppression test suggested poor adrenocorticotropic hormone (ACTH) response. She was diagnosed with OIAI. Magnetic resonance imaging showed a normal-sized pituitary gland without any focal lesions. She was administered systemic glucocorticoids and mineralocorticoids and prescribed a course of oral steroids. She was referred to substance use counseling [19].

## 6. Diagnosis

Since OIAI can result in significant morbidity, prompt recognition and treatment are optimal [20]. However, the presenting symptoms of OIAI tend to be vague and diffuse and might be readily attributed to the underlying condition rather than the opioid therapy [20]. Secondary adrenal insufficiency in general is challenging to diagnose [21,22]. Diagnostic delays are common, with an estimated six-month lag time between the onset of symptoms and clinical diagnosis [23]. In fact, 20% of OIAI patients wait over five years for an accurate diagnosis, and two-thirds (68%) are misdiagnosed before getting an accurate diagnosis [23]. It has been estimated that only about 10% of patients with OIAI will be diagnosed accurately [20]. Signs and symptoms of OIAI include musculoskeletal pain, weight loss, fatigue, gastrointestinal symptoms, and sexual dysfunction in patients with prolonged exposure to opioids [16,20].

There is no consensus definition for an OIAI diagnosis. Since OIAI is typically associated with a low level of cortisol, a low concentration of corticotropin, and a failure to launch an appropriate cortisol response when synthetic corticotropin is given, these tests are typically used. Confounding factors include the use of exogenous glucocorticosteroids or an abnormal pituitary gland. It is important to note that OIAI is caused by HPA-axis suppression rather than damaged or destroyed adrenal cortex regions. 

Although OIAI is a known clinical entity, it is not easily diagnosed and thus may not be adequately treated [2]. In fact, the optimal diagnostic workup for a patient on chronic opioid therapy suspected of having OIAI remains a subject of controversy, in part, because opioids can acutely suppress cortisol and may disrupt the circadian rhythm of cortisol production, which can affect the diagnostic tests [24,25]. 

A variety of diagnostic tests have been described in the literature and are summarized in Table 1 [18,21,24,25].

In healthy humans, serum glucocorticoid concentrations in the blood can change five-fold over the course of the day with the highest levels observed upon arising and the lowest during sleep; cortisol is also released in response to physical and emotional trauma apart from the circadian schedule [26]. When healthy subjects are administered intravenous morphine, their plasma cortisol levels will drop within 60 min. Cortisol production can also be thrown off by abnormal circadian rhythms, which are common among people with OUD [27,28]. For example, the morning cortisol test anticipates the subject will have peak daily cortisol at approximately 30 min to two hours after arising, but this only occurs in people with normal circadian rhythms, that is, it excludes most of the population with suspected OIAI [29].

The European Society of Endocrinology has recommended morning cortisol testing as a first-line diagnostic approach for adrenal insufficiency and recommends a cosyntropin stimulation test if the cortisol test is inconclusive [30]. However, this guidance relates to primary adrenal insufficiency and the guidelines do not specifically address how such recommendations apply to patients taking chronic opioid therapy. The cutoff values for cortisol levels can also vary. In many cases, 500 nmol/L is used. As might be suspected, when using a lower value, such as 405 nmol/L, fewer patients will be diagnosed with OIAI [25]. Measuring cortisol in the urine is not considered a helpful test, but it may be of value when considering the patient’s overall hormonal status [2].

A corticotropin (cosyntropin) stimulation test (CST) can be performed and is recommended if the cortisol test is inconclusive, but OIAI has been known to occur in patients with normal CST results [2]. From a clinical vantage point, it is presumed that people with OIAI will have below-normal levels of serum cortisol and a low corticotropin concentration; it is also thought that OIAI will cause the body to be unable to offer an appropriate cortisol response following the administration of corticotropin [2]. It should be noted that CST scores may be influenced by atrophy of the adrenal cortex which can be caused by primary adrenal insufficiency and/or an extended period of time without corticotropin [2]. Administration of synthetic corticotropin may stimulate the adrenal cortex to the degree it can provide a cortisol response of 18 to 20 µg/dL, but the exact cutoff values for OIAI diagnosis remain disputed and can vary, even across different hospitals and testing facilities [2].

Perhaps the gold standard diagnostic test for OIAI is the insulin tolerance test [13], but it is not always available at all locations. The cosyntropin stimulation test (CST) may be taken as a surrogate, because its results align with the insulin tolerance test [31]. Note that the CST has a sensitivity of 0.64 and specificity of 0.93, when the standard cutoff of 10 to 20 µg/dl is used [32]. The overnight test using metyrapone, a reversible 11β-mono-oxygenase inhibitor, may also be helpful [13]. Screening for basal cortisol levels should be followed by an insulin tolerance test, when available, if the cortisol level is low or inconclusive [33].

In a clinical study of 40 chronic opioid patients diagnosed with OIAI, patients took a mean of 105 MME/day (range 60 to 200 mg) for a median of 60 months upon inclusion into the study [20]. OIAI was diagnosed by low morning cortisol levels, baseline ACTH and/or DHEA-A, or an abnormal cosyntropin stimulation test [20].

Note that OIAI may present in unusual ways. The literature reports on the case of a 25-year-old man with OIAI presenting as hypercalcemia [34]. Following treatment for a critical illness that involved prolonged opioid exposure, he was diagnosed with hypercalcemia and treated with a three-day course of intravenous saline and glucocorticoid replacement. His hypercalcemia resolved, but his hypoadrenalism persisted until he discontinued opioids [34]. This case serves as an example of two concomitant and interacting endocrinopathies. Hypercalcemia can occur secondary to adrenal insufficiency and may be the presenting symptom [34].

## 7. The Clinical Course

The clinical course of OIAI has yet to be thoroughly elucidated [2]. A retrospective study of 40 OIAI patients conducted between 2006 and 2018 at a single center described the natural course of OIAI [20]. The population was 75% female with a median age of 49 years; the vast majority (95%) took opioids daily while 5% used opioids only as needed, with a median daily dose of 105 MME (range: 60 to 200 MME). The most frequently reported symptoms were fatigue (73%), musculoskeletal pain (53%), weight loss (43%), headache (30%), and nausea (20%). An adrenal crisis occurred in one patient during the study, but no patient presented with an adrenal crisis. A variety of laboratory values were used to diagnose OIAI and included low morning cortisol, baseline ACTH and/or DHEAS values, and CST testing. Median values of serum cortisol were 3 (range 1.4 to 5) µg/dL (normal > 7); ACTH was 9.7 (range 4.7 to 15) pg/mL (normal > 10); and DHEAS was 20 (range 15–32) µg/dL (normal > 50). Twenty-one percent were diagnosed with opioid-induced hypogonadism, which was more common in men. Most patients were prescribed hydrocortisone (95%) and of the subset of patients with follow-up data (n = 33), 42% showed symptomatic improvement [20].

## 8. Management of OIAI

The objective of OIAI treatment is the replication of the deficient cycles of physiologic cortisol, which should eliminate the OIAI symptoms. This seems reasonable in theory, but the currently marketed glucocorticoids are somewhat limited. The most commonly prescribed exogenous glucocorticoid, hydrocortisone, is typically administered in two or three daily divided doses for a total of 15 to 25 mg/dL [2]. Oral glucocorticoid medications can be used to regulate cortisol release, but the dose must be carefully determined to avoid excessive replacement [2]. Other glucocorticoid products, such as prednisone, prednisolone, and dexamethasone, offer the advantage of longer half-lives, but they are associated with increased glucocorticoid effects and may result in greater suppression of endogenous cortisol [2]. Dosing is based on symptomatic relief. Improved delivery systems and formulations for exogenous glucocorticosteroids are in development or new to market [2].

Complete discontinuation of opioids reverses OIAI, but the timeline for full recovery has not been conclusively established [2]. Note that long-term opioid therapy must be tapered rather that discontinued abruptly in order to mitigate or even avoid withdrawal symptoms [35]. It may be advisable for clinicians to monitor adrenal function during opioid tapering and after discontinuation [2].

Ideally, patients should be informed about the potential risks for adrenal insufficiency, hypogonadism, and other opioid-associated side effects before opioids are initiated [36]. Patients already taking chronic opioids should be periodically informed about these conditions and encouraged to report related symptoms which they may not necessarily associate with opioid use [36].

When opioid therapy must continue, OIAI can be managed if the patient can be educated and caregivers are willing to assist. When patients are under particular stress (major psychological stress, trauma, impending or recent surgical procedures), they should increase their intake of corticosteroids [2]. If the patient falls ill, the dose of corticosteroids should be doubled or even tripled for two or three days. If symptoms are too severe or oral intake is not possible, intramuscular corticosteroids may be used [2]. Patients with OIAI should wear a medical alert bracelet for emergency responders [2].

In a case report, a 46-year-old woman who had been treated for non-cancer pain for four years with a transdermal fentanyl patch (90 to 120 MME/day) developed fatigue, decreased energy, and loss of appetite in her second year of treatment [37]. Seeking treatment for worsening opioid-induced constipation and abdominal pain, a diagnosis of OIAI was reached based on her long-term opioid exposure, laboratory tests including the corticotropin-releasing hormone stimulation test, and the exclusion of other hypothalamic-pituitary disorders. Oral corticosteroid therapy was prescribed but did not relieve her constipation or abdominal pain; opioid tapering, rotation, or discontinuation was not possible because of her painful symptoms and anxiety [37].

## 9. Adrenal Insufficiency in the COVID Era

There is some controversy as to whether patients with primary or secondary adrenal insufficiency, including OIAI, have an elevated risk of contracting COVID-19 and face a greater risk of a complicated course, because the infection may trigger a possibly life-threatening adrenal crisis [38,39,40,41]. In cases of a patient with both COVID-19 and OIAI, an increased dose of glucocorticoid replacement therapy is recommended [38]. Other autoimmune endocrinopathies have been reported in case studies associated with COVID-19 infection and certainly adrenal dysfunction has occurred in other infectious diseases [42,43].

An interesting academic speculation is whether so-called “long COVID” is actually an adrenal problem that emerges after the acute COVID-19 symptoms resolve [43]. The symptoms of secondary adrenal insufficiency and long COVID overlap somewhat, leading to the hypothesis that long COVID may be COVID-induced adrenal insufficiency [43]. Viral tropism to adrenal glands is not unusual in viral infections, and adrenal insufficiency has been reported in patients with meningococcal sepsis and tuberculosis adrenalitis [44]. Autopsies of COVID-19 patients have reported both fibrin and microthrombi deposits in the adrenal capillaries [45]. Adrenal insufficiency in the aftermath of acute COVID infection would likely require several weeks to develop [43]. It goes beyond the scope of this review of OIAI to explore the connections between COVID-19 and adrenal insufficiency, but care should be taken with OIAI patients who contract COVID-19.

## 10. Adrenal Androgens

Adrenal androgens contribute far more to the overall androgen stores in females than in males [2]. The serum levels of the weak adrenal androgen, dehydroepiandrosterone sulfate (DHEA-S), are lower in long-term opioid users compared to those who do not use opioids [46]. The effect of DHEA-S supplementation in patients with OIAI has not been well studied and is not generally recommended, but remains an area for investigation [2]. For young women with OIAI and receiving optimal glucocorticoid replacement therapy, DHEA-S supplementation is sometimes considered a means to treat symptoms of diminished libido, depression, and low energy [2].

## 11. Adrenal Crisis

Adrenal crisis is a potentially lethal acute emergency that may occur in primary or secondary adrenal insufficiency, including OIAI. Adrenal crisis differs from adrenal insufficiency, although the term acute adrenal insufficiency is sometimes used interchangeably with adrenal crisis [10]. In some instances, an adrenal crisis is the initial presentation of patients with OIAI or other forms of adrenal insufficiency. There is no expert consensus definition of adrenal crisis, but the condition is characterized by hypotension that can be treated effectively with parenteral glucocorticoids [10]. Other symptoms may mirror those that occur with OIAI or adrenal insufficiency, such as confusion and abdominal pain [10]. There is no clear-cut evidence as to what can trigger an adrenal crisis or why crises occur in some patients but not others [10]. Infection, chemotherapy or immunotherapy, and nonadherence to glucocorticoid regimens in patients with adrenal insufficiency have been implicated in an adrenal crisis [10].

In addition to the symptoms of OIAI, patients in an adrenal crisis may suffer postural hypotension, nausea, vomiting, myalgia, and cardiovascular effects, even escalating to cardiovascular collapse [2]. Treatment should be hydration resuscitation (1 L of 0.9% saline over 1 h followed by infusion as needed), parenteral glucocorticosteroids (100 mg bolus dose of IV hydrocortisone, then 200 to 300 mg of hydrocortisone per day) [10,47]. Few studies of an adrenal crisis have assessed patients for OIAI [13].

## 12. Clinical Considerations for Daily Practice

All patients treated with long-term opioid therapy should be assessed for OIAI, particularly those who experience symptoms of weight loss, fatigue, lethargy, and nausea [13]. Secondary adrenal insufficiency is associated with diminished work capacity and reduced quality of life [8]. Unfortunately, fatigue, myalgia, and gastrointestinal symptoms are very common among chronic pain patients regardless of the presence of OIAI [48,49]. This is probably the main reason why this condition has long been unknown among physicians and is still underdiagnosed in current clinical practice. Moreover, glucocorticoids are often part of the analgesic plan, particularly among cancer patients with bone metastases [50]; therefore, OIAI is pharmacologically masked and reveals no clinical signs or symptoms. Postural symptoms suggestive of an adrenal disorder should also be assessed but they are not as frequent because OIAI does not affect aldosterone secretion [9]. Laboratory testing should be conducted and overall management should include glucocorticoid replacement and, ideally, the reduction or discontinuation of opioid therapy [13]. Since MME was found to be the sole predictor for the development of OIAI in one prospective study [14], it is recommended that patients taking ≥ 20 MME/day be monitored for potential OIAI [14].

Unlike other-induced endocrinopathies [51], clinical awareness of OIAI is limited. In a cross-sectional survey of internal medicine healthcare professionals who care for patients in chronic opioid therapy (n = 91 respondents out of 300), 19% stated they felt comfortable in their knowledge of opioid-associated side effects and 68% of non-endocrine specialists recognized OIAI as an opioid-induced endocrinopathy [33]. Significantly more endocrinologists (38%) correctly identified the symptoms of OIAI compared to other healthcare providers in the survey (9%), *p* < 0.01. The majority of respondents thought online resources, continuing education activities, and other training interventions could improve knowledge about how to diagnose and treat OIAI [33].

Despite limited awareness of OIAI, this form of central adrenal insufficiency is more prevalent than most clinicians appreciate. There is no consensus as to diagnostic testing, but frequently used are a combination of a morning serum cortisol test, ACTH test, and CST stimulation test. Note that cortisol tests at random times other than morning have less clinical utility for OIAI diagnosis [18]. Low ACTH suggests a central etiology and the CST stimulation test suggests a need for glucocorticoid replacement therapy. High doses of opioids can suppress the HPA axis and cause hypoadrenalism, but the adrenal glands may retain their ability to respond to corticotropin; sometimes normal or near-normal CST stimulation tests occur in patients with OIAI [18].

Central adrenal insufficiency usually does not reduce aldosterone levels, conversely to primary adrenal in-sufficiency, because aldosterone secretion by the adrenal glands is mainly regulated by the renin-angiotensin-aldosterone system [9]. See Figure 2. This is the reason why OIAI usually does not affect plasma levels of sodium and potassium and why it does not alter blood volume or blood pressure. Cardiovascular effects become clinically evident only during adrenal crisis, when they add to other OIAI symptoms, leading to a potentially life-threatening condition that may necessitate resuscitation [8]. See Figure 3.

Central adrenal insufficiency may be caused by exogenous steroids, pituitary tumors, infiltrative diseases, head trauma, and apoplexy, so these must be ruled out in any diagnosis [18]. Cessation of opioid therapy resolves the condition [52], but this is not possible for all patients. When complete discontinuation cannot be achieved, the lowest effective dose of opioid should be used [53]. Opioid-tolerant patients should be tapered to avoid or reduce withdrawal [54].

While there are two case studies describing tramadol-induced adrenal insufficiency, differences between weak and strong opioids with respect to OIAI remain to be elucidated [52,55]. However, regarding other endocrine effects of opioids, such as OPIAD syndrome, opioids with reduced mu load, such as tapentadol, have been shown to have a lower impact on sex hormone concentrations compared with strong opioids, such as oxycodone and morphine [56]. Therefore, further studies are warranted to elucidate the clinical impact of different opioids on OIAI.

## 13. Experimental Evidence of the Peripheral Effects of Opioids

The majority of reported studies regarding the effects of opioids focus on the central nervous system. However, experimental evidence, mostly derived from studies of animal models, indicates that the expression of opioid receptors is not restricted to the central nervous system, [57] suggesting that opioid-induced effects can also be elicited by activation of these receptors outside of the CNS, in agreement with previous observations [58,59,60]. In fact, besides the classical localization in the central nervous system, they can also be detected both at the mRNA and protein levels in a wide variety of peripheral tissues of most species investigated so far, including humans. Specifically, they have been detected in the peripheral sensory neurons, where their activation exerts analgesic effects without inducing centrally mediated side effects [61]. They have been identified in testicular germ cells, where they are actively involved in spermatogenesis [62]. In the heart, they are thought to play a protective role in heart failure [63]. In the gastrointestinal system [64], they play a role in the regulation of gastrointestinal fluid secretion and gut motility [65]. In the immune system, they modulate immune response [66,67]. In the synovium of patients with joint trauma, osteoarthritis, and rheumatoid arthritis, it is believed they act locally to counteract pain and inflammation [68]. Last but not least, they are found in the mononuclear immune cells located inside and surrounding cancer tissues, where they appear to mediate a thermal hypoalgesic response [69,70]. An extensive list of mRNA distribution of the three opioid receptors (mu-1OPRM1, kappa 1-OPRK1, and delta 1 OPRD1) has been published on the Human Protein Atlas website and elsewhere [71,72]. The mRNA expressions of the three opioid receptor subtypes in human tissues appear in Figure 4.

The expression of the genes coding for the three major subtypes of opioid receptors, namely the MOR, DOR, and KOR, was detected at high levels in the adrenal glands of rats [73], sheep [74], mice [75], pigs [76,77], bovines [78], guinea pigs [79], and humans [57].

The presence of these opioid receptors in the adrenal glands is consistent with a possible direct role of the opioids in modulating adrenal gland function. In addition, enkephalin and enkephalin-like peptides have been detected in the adrenal glands of different species with much higher quantities in bovine, canine, and human adrenals than in guinea pig or rat adrenals [80,81,82]. Relatively high levels of enkephalin have also been found in the adrenal medulla, contained within the same storage vesicles as catecholamines [83].

In this regard, experiments performed on an intact perfused rate adrenal preparation showed that endogenous opioid peptides may stimulate aldosterone and corticosterone secretion [84,85] and modulate adrenal vascular tone [86]. This experimental evidence was confirmed by other authors, who demonstrated that all of the opioid peptides tested, namely the β-endorphin, the Leu-enkephalin, the Met-enkephalin, and its long-acting analog D-ala2-met5-enkephalinamide (DALA) were capable of exerting a stimulatory effect on aldosterone secretion by cells of the zona glomerulosa, the most superficial layer of the adrenal cortex. With the exception of Leu-enkephalin, all of these were able to stimulate corticosterone secretion by the cells of the inner fasciculata/reticularis zones as well [87]. These stimulatory effects did not depend on or correlate to the stimulatory effects of ACTH. The effects of opioid peptides in peripheral tissues appear to be specifically mediated by the different opioid receptors, with the MOR able to mediate such effects on the cells of both the zona glomerulosa and inner zones, presumably through Ca2+ influx and activation of phospholipase C [88].

Evidence in support of the direct effects of opioids on the adrenal glands, independent from their effect on the HPA axis, has been reported in humans. Since such studies are complicated by the inevitable influence of the central effects on the central nervous system and, in particular, on the hypothalamus, a group of patients with hypothalamic-pituitary disconnection due to various pathological conditions (including craniopharyngioma, chordoma, suprasellar meningioma, or pituitary macroadenoma) was enrolled in a study [89]. In these patients, the administration of naloxone increased circulating cortisol but not ACTH levels when compared to control patients administered saline solution. This suggests that naloxone might exert a direct effect on cortisol secretion at the adrenal gland level. Another possible way to demonstrate the direct effect of opioids on adrenal glands is to isolate human adrenocortical cells [90]. In such a cell model, the β-endorphin stimulation exerted a steroidogenic effect, especially on the production of androstenedione and DHEA, greater than that observed after stimulation with ACTH.

The full physiological role of opioid peptides in the periphery remains to be elucidated. The presence of opioid receptors in the adrenal glands suggests a paracrine or autocrine role in adrenal function [57]. A more in-depth knowledge of the specific expression of opioid receptors and their ligands in the adrenal glands—both of which control stress, mood, pain, and inflammation—may allow for the design and development of peripherally restricted ligands that could offer analgesic benefits while avoiding the centrally mediated adverse effects of opioids.

## 14. Discussion

Opioids are used widely in the developed world, both by pain patients and recreational drug users, two distinct populations with some overlap. While many opioid-associated side effects, such as opioid-induced constipation, lethargy, nausea, vomiting, dizziness, and pruritus are well known, opioid-associated endocrinopathies are underappreciated [13]. In addition to those who take opioids medically or recreationally, OIAI may also occur in those on medication-assisted opioid therapy, that is, the population using methadone or buprenorphine in their recovery and rehabilitation. Although this represents a substantial and growing population of those who have prolonged exposure to opioids, there are almost no studies of the functional effects of such treatments [91]. The lack of a consensus definition of OIAI (or any opioid-associated endocrinopathy) has led to lack of awareness of the condition and no specific guidance as to how it should be diagnosed or treated.

While opioid mortality is frequently discussed in scientific literature as well as in the news media, opioid morbidity is less often discussed. Like OIAI, opioid morbidity remains a term that most clinicians understand but few can diagnose. Opioid morbidity would likely encompass conditions such as opioid-induced hyperalgesia [92], endocrinopathies [93], and disorders of bone metabolism associated with opioid use [94]. These are important topics that are not often the topic of investigation or inquiry. Mental disorders also occur concomitantly with opioid use disorder, but it is not clear if mental health conditions might precipitate opioid use or result from it. Since opioid clinical trials typically exclude patients with diagnosed mental health conditions, there is not a lot of evidence for clinicians to understand how opioids, particularly long-term use of recreational opioids, can affect such individuals [95]. Further study is urgently needed to better understand opioid-associated endocrinopathies and how to treat them.

## 15. Conclusions

OIAI is an under-recognized condition of adrenal insufficiency secondary to long-term exposure to opioids. OIAI can cause symptoms, and may result in potentially life-threatening adrenal crises, but can be managed. This management may be crucial for that subset of chronic opioid therapy patients who cannot or do not see the benefit of reducing or ceasing opioid treatment. Understanding how to diagnose and treat OIAI is crucial, particularly since opioids are widely used. Further controlled clinical studies are warranted to better analyze the central versus peripheral opioid effects on adrenal gland hormone production.

## Figures and Tables

**Figure 1 ijms-24-04575-f001:**
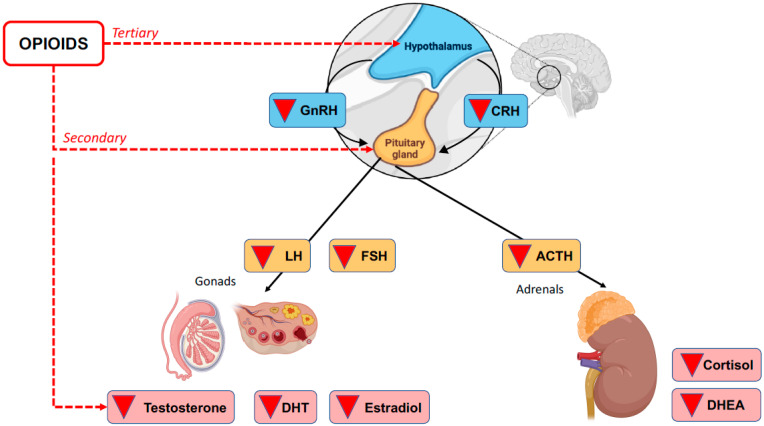
Opioid-induced secondary and tertiary adrenal insufficiency.

**Figure 2 ijms-24-04575-f002:**
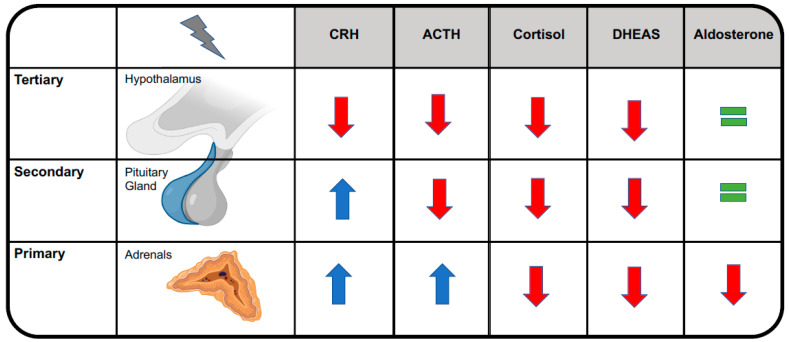
Distinguishing among primary, secondary, and tertiary forms of adrenal insufficiency. ACTH, adrenocorticotropic hormone; CRH, corticotropin-releasing hormone; DHEAS, dehydroepiandrosterone sulfate.

**Figure 3 ijms-24-04575-f003:**
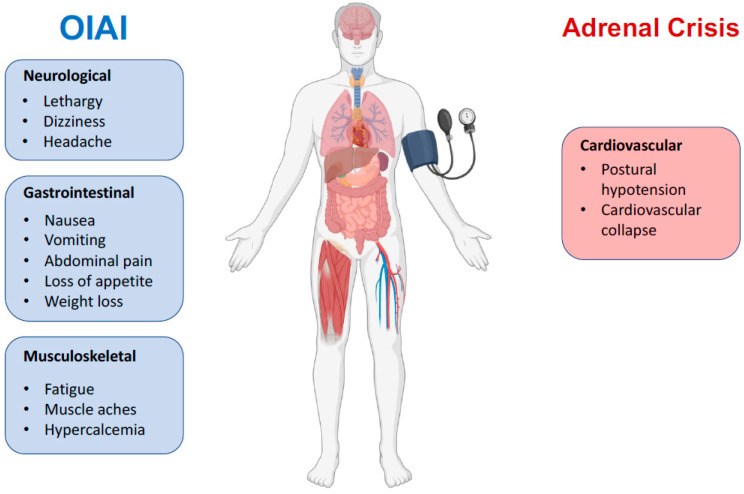
Key clinical features of OIAI versus adrenal crisis. Opioids usually do not cause cardiovascular symptoms except in the context of adrenal crisis.

**Figure 4 ijms-24-04575-f004:**
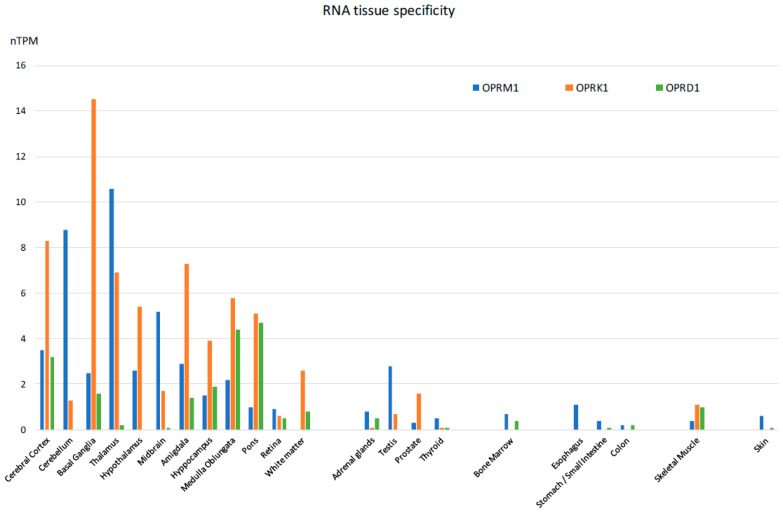
Tissue-specific RNA expression of the three opioid receptors in humans [Adapted from [71]].

**Table 1 ijms-24-04575-t001:** Diagnostic testing strategies.

Priority	Test	Values	Considerations
1	Morning cortisol	<3 to 3.6 µg/dL is suggestive	Cortisol secretion has a circadian pattern and can be difficult to assess. Values after 8 a.m. are less likely to be reliable.
2	DHEAS	>54.5 µg/dL	Ranges vary by sex and age; low-normal values can be suggestive.
3	CST test	<500 nmol/L	
4	Insulin tolerance test or ACTH stimulation test	Cutoff values not entirely established	Applicable when first two tests are inconclusive or there remain doubts. An abnormal ACTH is highly specific for adrenal insufficiency. Low ACTH values indicate central etiology.
Others	Metyrapone stimulation test Glucagon stimulation test		Note that these tests are not always highly specific.

ACTH, adrenocorticotropic hormone; CST, cosyntropin stimulation test; DHEAS, dehydroepiandrosterone sulfate.

## Data Availability

Not applicable.

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
