# Peer review of "A Closer Look at Opioid-Induced Adrenal Insufficiency: A Narrative Review"

_ijms, 2023, doi:10.3390/ijms24054575_

Round 1

Reviewer 1 Report

Very well written and adds to quality of literature.

Exposure should be one word- not hyphenated

Minor comments-comprehensive overview

address if HPA alterations may be more adverse if opioids used prior to puberty- since it is mentioned in terms of development

The exact mechanisms by which long COVID can lead to adrenal insufficiency are not yet clear, but it may be related to inflammation and damage to the adrenal glands caused by the viral infection. Indeed beyond scope to integrate a mechanistic exploration, yet linking to other sections may serve as hypothesis generating. while there have been reported cases of adrenal insufficiency in individuals with long COVID, it is still a relatively rare occurrence offering opportunity for investigation

Indeed, . Mental disorders also occur concomitantly with opioid use disorder but it is not clear if mental health conditions might precipitate opioid use or result from it. Since opioid clinical trials typically exclude patients with diagnosed mental health conditions, there is not a lot of evidence for clinicians to understand how opioids, particularly long-term use of recreational opioids, can affect such individuals." introduction of this construct doe snot fit in the conclusion.

Reviewer 2 Report

This review concentrates on opioid-induced adrenal insufficiency (OIAI), a potentially life-thretening condition. A long-term exposure to opioids in both humans and animals has been associated with suppression of the HPA axis. Given increasing numbers of patients having a chronic prescription of opioids, the syndrome takes on importance in developed countries.  The ethiopathology of OIAI and the role of HPA axis, as well as treatment modalities are clearly described in this paper. 

To do this, the Authors properly searched PubMed and Cochrane databases showing only scarce available information published on OIAI. They based, however their conclusions on a total 95 publications of supporting literature. Therefore, this review is an updated and comprehensive summary of actual knowledge on adrenal insufficiency in opioid patients and on a proper treatment, based on well selected studies.

The 4 figures and the table are correct and informative. The English is perfect.

I recommend publishing this paper for the interest of IJMS readers.